# In silico functional annotation of hypothetical proteins from the *Bacillus paralicheniformis* strain Bac84 reveals proteins with biotechnological potentials and adaptational functions to extreme environments

**Md. Atikur Rahman**[1], **Uzma Habiba Heme**[2], **Md. Anowar Khasru Parvez**[3]*

**1** Institute of Microbiology, Friedrich Schiller University Jena, Thuringia, Germany, **2** Faculty of Biological Sciences, Friedrich Schiller University Jena, Thuringia, Germany, **3** Department of Microbiology, Jahangirnagar University, Savar, Dhaka, Bangladesh

* khasru73@juniv.edu

## Abstract

Members of the *Bacillus* genus are industrial cell factories due to their capacity to secrete significant quantities of biomolecules with industrial applications. The *Bacillus paralicheniformis* strain Bac84 was isolated from the Red Sea and it shares a close evolutionary relationship with *Bacillus licheniformis*. However, a significant number of proteins in its genome are annotated as functionally uncharacterized hypothetical proteins. Investigating these proteins' functions may help us better understand how bacteria survive extreme environmental conditions and to find novel targets for biotechnological applications. Therefore, the purpose of our research was to functionally annotate the hypothetical proteins from the genome of *B. paralicheniformis* strain Bac84. We employed a structured in-silico approach incorporating numerous bioinformatics tools and databases for functional annotation, physicochemical characterization, subcellular localization, protein-protein interactions, and three-dimensional structure determination. Sequences of 414 hypothetical proteins were evaluated and we were able to successfully attribute a function to 37 hypothetical proteins. Moreover, we performed receiver operating characteristic analysis to assess the performance of various tools used in this present study. We identified 12 proteins having significant adaptational roles to unfavorable environments such as sporulation, formation of biofilm, motility, regulation of transcription, etc. Additionally, 8 proteins were predicted with biotechnological potentials such as coenzyme A biosynthesis, phenylalanine biosynthesis, rare-sugars biosynthesis, antibiotic biosynthesis, bioremediation, and others. Evaluation of the performance of the tools showed an accuracy of 98% which represented the rationality of the tools used. This work shows that this annotation strategy will make the functional characterization of unknown proteins easier and can find the target for further investigation. The knowledge of these hypothetical proteins' potential functions aids *B. paralicheniformis* strain Bac84 in effectively creating a new biotechnological target. In addition, the results may also facilitate a better understanding of the survival mechanisms in harsh environmental conditions.

**Data Availability Statement:** All relevant data are within the paper and its Supporting Information files.

**Funding:** The author(s) received no specific funding for this work.

**Competing interests:** The authors have declared that no competing interests exist.

## Introduction

*Bacillus paralicheniformis* is a newly discovered species in the *Bacillus* genus [1]. It is phylogenetically closely related to *B. licheniformis* [1, 2]. In the biotechnology sector, *B. licheniformis* has already been employed to produce biochemicals, enzymes, antibiotics, and other products [1, 3]. Several current investigations have indicated that *B. paralicheniformis* species have a strong potential for the biosynthesis of antimicrobial compounds [4, 5]. One of the strains can also inhibit plant pathogenic microbes [6]. In this way, *B. paralicheniformis* may be of biotechnological relevance but still, it has remained largely unexplored.

*B. paralicheniformis* is a gram-positive, facultatively anaerobic, rod-shaped, motile, and endospore-forming *Bacillus* species [1]. The *B. paralicheniformis* strains are found in a variety of habitats, including soil, freshwater, marine, and niches associated with food [1, 4, 6]. This strain is adapted to survive in extreme conditions such as high osmolarity which provides it with metabolic capabilities similar to industrial strains [4]. The *B. paralicheniformis* strain Bac84 was isolated from the Red Sea which is an ecosystem of harsh, extremely saline, and high temperature [4]. Hence, this strain may be a potential microbial cell factory to produce both thermo-tolerant and osmotolerant enzymes that may be more suitable for use in industry as well as able to survive frequent exposure to these extreme conditions [7]. This particular strain showed promising antibacterial activity against three-indicator pathogens: *Salmonella typhimurium*, *Staphylococcus aureus*, and *Pseudomonas syringae* [8]. Additionally, one very closely related strain (*B. paralicheniformis* Strain GSFE7- 95% genome sequence similarity) has been reported to be involved in the promotion of halotolerant plant growth [9]. Besides, another closely related strain (*B. paralicheniformis* Strain CCMM B940 which shares 98.94% identity with *B. paralicheniformis* strain Bac84) can break down complex polysaccharides [10].

The genome of *B. paralicheniformis* strain Bac84 has been fully sequenced and published [4]. According to the National Center for Biotechnology Information database—NCBI repository, it encodes 4,237 proteins (CP023665.1). However, 414 coding sequences have been anticipated to encode for proteins without any expression and function-associated data. These sequences have been assigned as "hypothetical". These hypothetical proteins (HPs) have constituted a considerable portion (9.8% of the total number of proteins) of the genome. Functional annotation is necessary for these HPs to find the possible roles in the cell which can lead to an understanding of new structures, and functions in this bacterium. Several studies have revealed the expression of HPs [11–13]. Homology-based gene annotation has been assigned previously to predict the unknown functions of numerous HPs in several organisms [14–18]. Additionally, numerous bioinformatics tools are available to determine the functions of the HPs such as Pfam, InterPro, CATH, SUPERFAMILY, SMART, CDD-BLAST SCANPROSITE, and many more [17–23]. Moreover, the STRING database is also an essential way of protein-protein interaction (PPI) determination to understand the protein functions in a biological network [24–26]. Hence, the PPI study of these HPs can lead to inferences about their biological functions [27]. Furthermore, the tertiary structure modeling through homology searches utilizing the SWISS-MODEL server is important to find the function of unknown proteins [28].

In this study, we aimed to determine the functional roles of the HPs from the *B. paralicheniformis* strain Bac84. We utilized an annotation-based workflow to determine the functions of the HPs for the identification of new biotechnologically important proteins as well as novel proteins contributing to the survival of this bacterium in extreme environments. We successfully identified potential target proteins in the *B. paralicheniformis* strain Bac84. It may eventually be possible to develop new biotechnological applications based on further experimental validation of these identified proteins.

## Materials and methods

### Sequence retrieval

The genome of *B. paralicheniformis* strain Bac84 was used (CP023665.1). It has 4,376,831 bp in length containing 4413 genes. It encodes 4,237 proteins and 414 are HPs among those (https://www.ncbi.nlm.nih.gov/genome/). The HPs' sequences were obtained in FASTA format for the analyses (S1 Table).

### Functional annotation of hypothetical proteins

Functional annotation was applied to the HPs to reveal their functions (Fig 1). Firstly, several publicly available tools and databases (Pfam, InterPro, CATH, SUPERFAMILY, SMART, SCANPROSITE, and CDD-BLAST) are listed in the S2 Table were used. These bioinformatics tools and databases assist to find the conserved domains and afterward categorize the proteins. Pfam [29], InterPro [30], SUPERFAMILY [20], and SCANPROSITE [31] were employed to interpret the functional roles of the HPs based on similarity. Additionally, SMART and CATH were used to search for functions of our HPs based on the domain architecture and to categorize the domains within the structural hierarchy respectively [32, 33]. Conserved Domain Database (CDD) was utilized to search conserved domains [34]. All these analyses were performed in the default parameters and the results are given in detail in the S3 Table. These web tools showed distinctive results and to perform downstream analyses, 37 HPs were filtered as these HPs exhibited functional domains or motifs in at least three of the bioinformatic tools (S4 Table).

We also have predicted the gene ontology of all the HPs using Argot[2.5] (Annotation Retrieval of Genel Ontology Terms) [35] (S5 Table) and the findings are illustrated in Fig 2.

We further used the FASTA sequences of the selected 37 HPs for manual annotation utilizing the Basic Local Alignment Search Tool (BLAST) [36]. Here, the NCBI nonredundant database and hits with an identity ≥ 90% were employed (S6 Table).

In addition, we used BPROM (in the default settings) to perform the promoter analysis of the 37 proteins [37]. All the DNA sequences were downloaded from the NCBI database. The Shine Dalgarno (SD) sequence was manually assigned in this case.

The DEG database was utilized to detect the essential genes with the screened 37 HPs [38]. The search was performed against the available genomes of *Bacillus subtilis 168*, and *Bacillus thuringiensis BMB171* in the default parameters (S7 Table).

### Prediction of physicochemical parameters and the Sub-cellular localization

The physicochemical parameters of the selected 37 HPs were theoretically measured using Expasy's Protparam server [39]. The predicted properties such as molecular mass, isoelectric point (pI), extinction coefficient, the total number of +/- residues, extinction coefficient, instability index, aliphatic index, and grand average of hydropathicity (GRAVY) were determined.

Determination of the protein cellular localization helps to estimate its function. In this study, PSORTb [40] and CELLO [41] were used to identify the proteins' location in the cell. PSORTb includes both lab experimental data sets as well as in silico predictions. In contrast, CELLO employs a two-level support vector machine (SVM) based system.

Furthermore, SOSUI [42], HMMTOP [43], TMHMM [44], and SignalP [45] were utilized to predict the transmembrane helices as well as determine the presence of signal peptide cleavage sites. All the results of these characterization analyses were listed in the S8 Table.

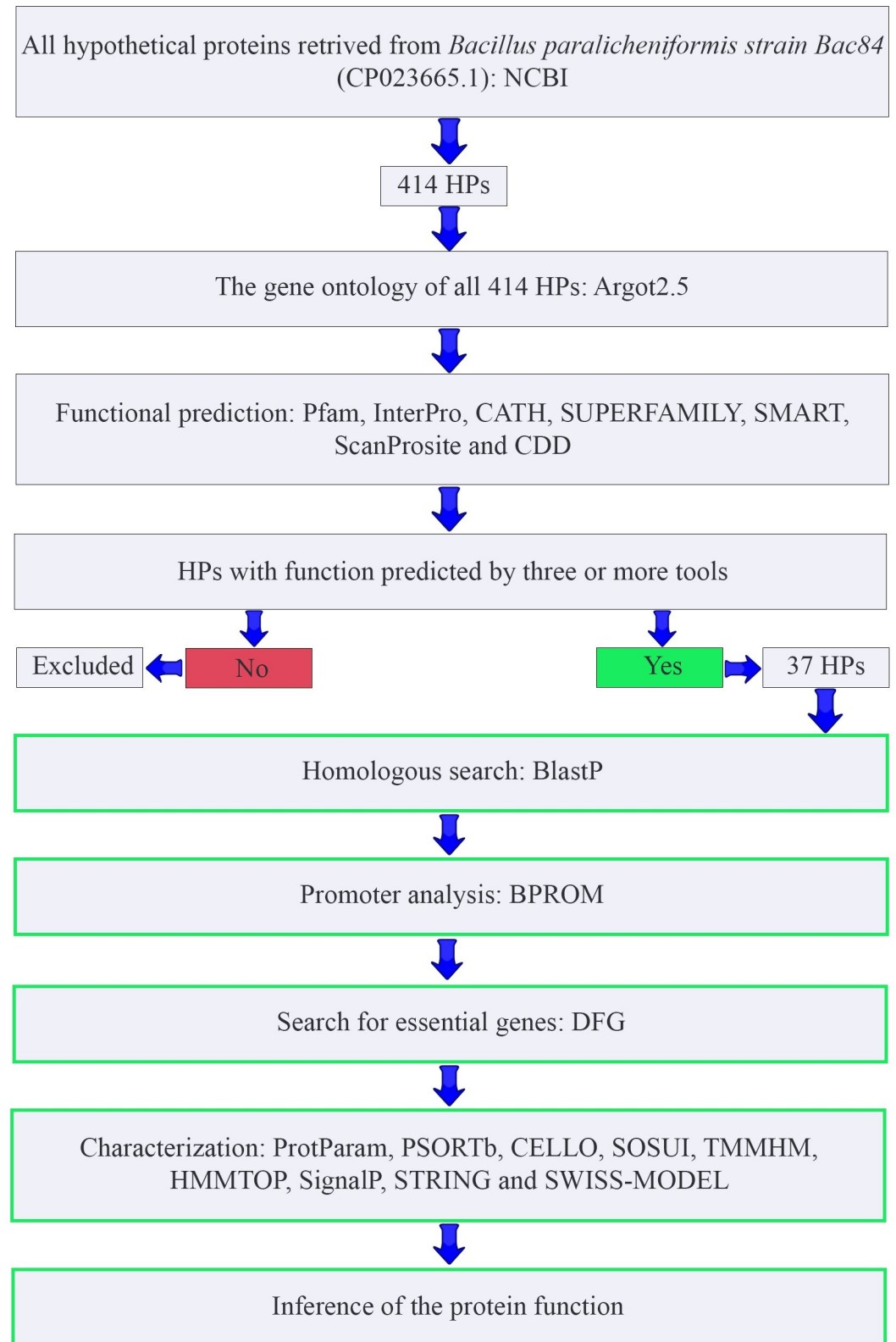

**Fig 1. Workflow representing the overall design of the study.** The tasks listed in the green outlined boxes were applied only after the analyzed HPs showed the same function in at least three different bioinformatics tools.

**(A) HPs distribution among the GO categories**

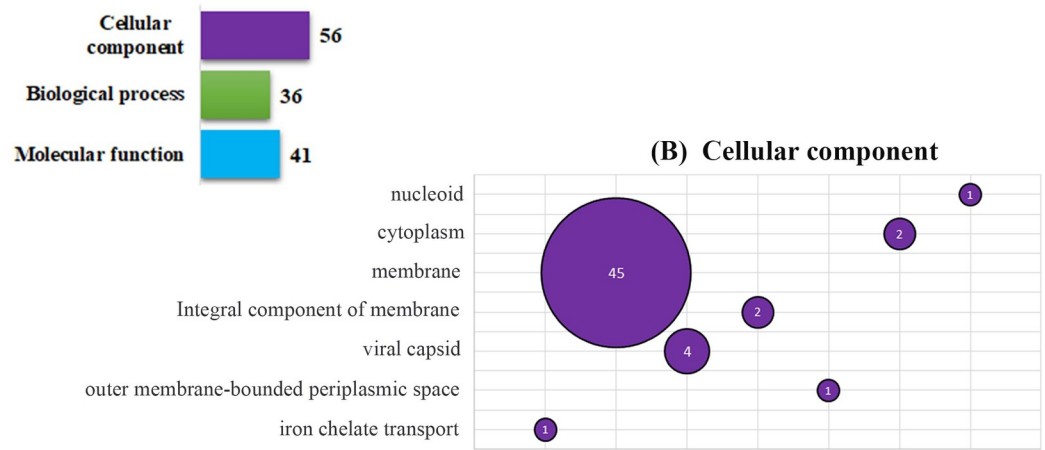

**(B) Cellular component**

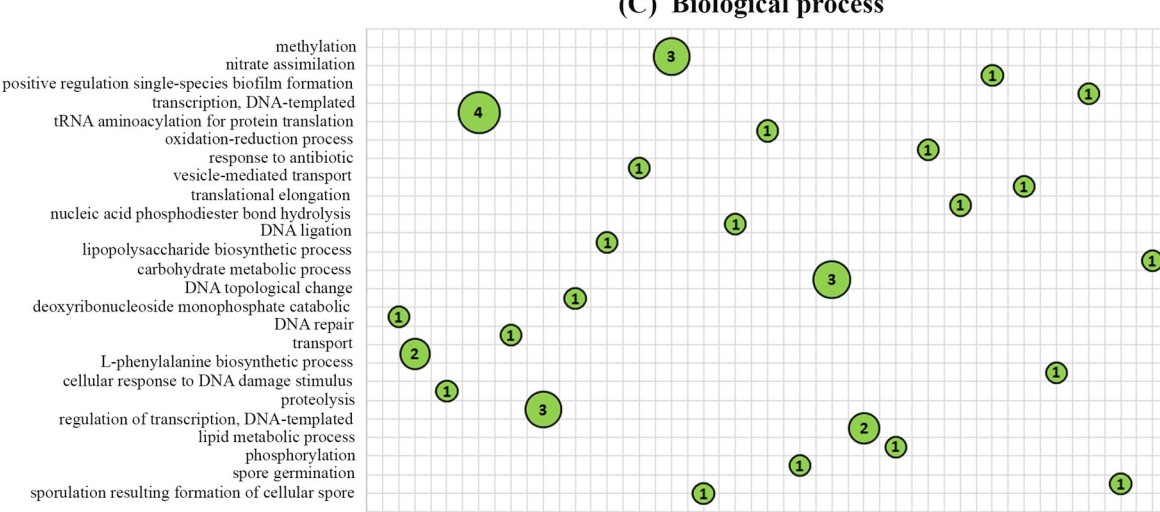

**(C) Biological process**

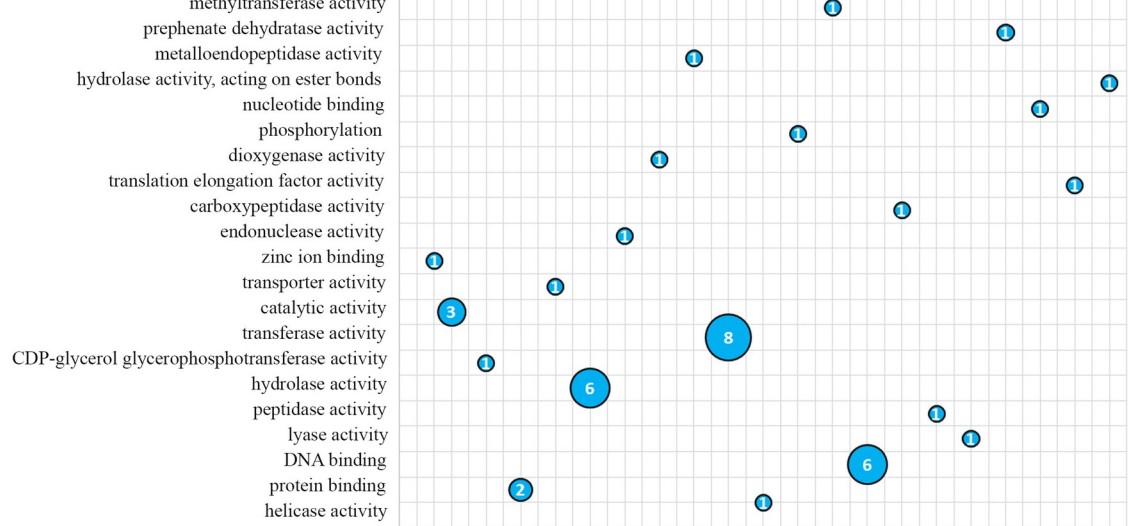

**(D) Molecular function**

**Fig 2. The gene ontology of all the 414 HPs.** (A) The distribution of the HPs among the three gene ontology categories. (B) Graph of the cellular components. (C) Graph of the biological processes. (D) Graph of the molecular functions. Here, the distribution of GO terms is presented on the Y axis and the area of the bubbles is relative to the number of proteins found in each category.

## Protein-protein interaction analysis

In this study, STRING software [24, 26] was used to predict interactive partners using a confidence score above 0.7 for ensuring the dependability of the predictions (S9 Table). We had to use the *Bacillus licheniformis* DSM 13 reference genome to generate the interaction networks as the dataset for any strain of *B. paralicheniformis* has not been available yet. Both the physical and functional associations were applied to compute the networks. The Cytoscape was used to visualize the interaction networks (S1 Fig).

## Tertiary structure prediction

Tertiary protein structures give significant insights into the molecular basis of protein function [46]. We used the SWISS-MODEL server [28] for homology modeling of the target proteins where only templates with an identity ≥ 30% were considered (S10 Table). The UCSF Chimera-1.16 was used to visualize the 3D structures as well as to perform the structural alignments (Fig 3A & 3B). Additionally, several predicted structures were also compared with the AlphaFold models to validate the structures.

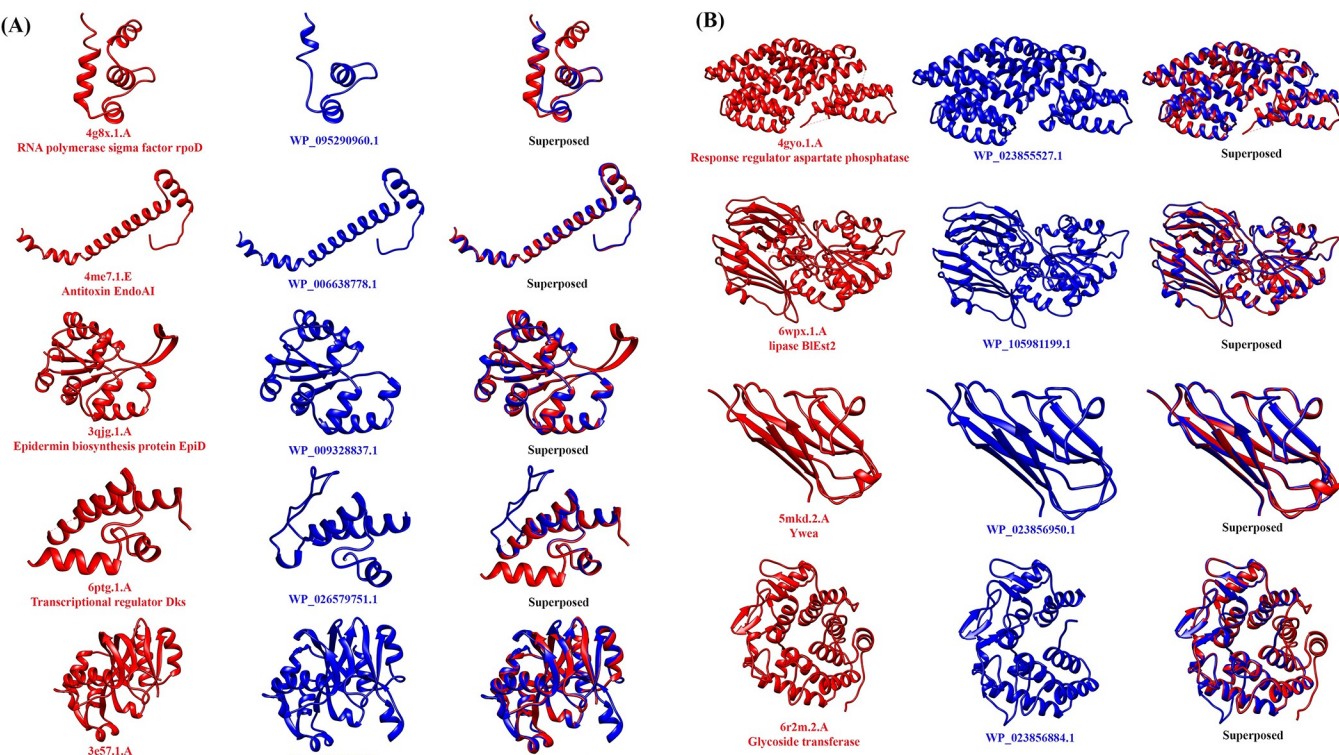

**Fig 3.** A & B. Tertiary structures analysis. Three-dimensional structures were modeled by the SWISS-MODEL server reliably using the templates with higher coverage, more than 30% of identity, and higher GMQE scores along with Ramachandran Favored percentages ≥90%. Only the templates determined by the X-ray crystallography with high resolution were used. The known proteins and the modeled structures are indicated in red and blue colors respectively. The proteins are orientated using the Chimera MatchMaker according to the optimal superposition of the matching residues.

## Performance assessment

We performed a ROC- receiver operating characteristic analysis with 100 functionally characterized proteins (S11 Table) from the genome of the *B. paralicheniformis* strain Bac84 to check the accuracy of the anticipated functions of our studied HPs [47]. These proteins were functionally checked using the seven databases used for our studied HPs.

For the interpretation, the binary numerals "1" and "0" were applied as the true positive and true negative respectively. The integers '2', '3', '4', and '5' were used to assess the prediction efficacy. After that, these datasets were submitted to the Web-based Calculator and calculated the specificity, sensitivity, accuracy, and the ROC area of each tool employed earlier for functional prediction of the HPs.

# Results and discussion

## Analysis of The hypothetical proteins from the *B. Paralicheniformis* strain Bac84 genome

DNA sequencing technologies are advancing, and high throughput sequencing technologies have allowed a significant number of bacterial genome sequencing. Sequence homology techniques are commonly used for the annotation of genes [48]. Nevertheless, these homology techniques alone are not always able to predict functions accurately and lead to false annotations [49]. Hence, multiple bioinformatic tools must be employed to assign functional annotations of HPs. In this study, we applied a number of effective tools and databases to do the annotation of HPs from the *B. paralicheniformis* strain Bac84.

We first identified the domains of the HPs which are structural, functional, and evolutionary parts of a protein, therefore providing the functional role of a protein [50]. We extensively analyzed all the 414 HPs sequences using Pfam, InterPro, CATH, SUPERFAMILY, SMART, SCANPROSITE, and CDD-BLAST (S3 Table). The results were evaluated aiming to assign functions to HPs and it revealed 37 HPs which demonstrated similar functions from three or more programs listed in Table 1. In this way, functional annotations were assigned with strong confidence to the HPs. For the rest HPs (n = 377), domains were recognized from less than three mentioned bioinformatic tools which are needed further assessments.

Further, the GO terms were determined using the ARGOT[2.5] server [35] that provides results based on the confidence scores. 133 HPs have GO term predictions among the 414 targets and the distribution among the GO categories was depicted in Fig 2. The rest of the HPs with no GO terms can be found in the S5 Table. Among the three categories, the largest cluster was cellular components followed by molecular functions and biological processes. We found seven different GO terminologies in the cellular component category including 45 having membrane function (Fig 2B). Although studying membrane proteins is difficult, it is well known that many membrane proteins play important roles in gram-positive bacteria's physiology [51, 52]. The membrane proteins come first in the interaction among cells and the environmental stresses [53]. These membrane HPs need to be analyzed as these may have considerable roles in the survival mechanism of the *B. paralicheniformis* strain Bac84 in extreme environments. For biological processes, twenty-five different GO terminologies were identified, mostly associated with transcription and DNA-related processes (Fig 2C). Transcriptional regulation is a crucial process for a living organism. The cell can respond to intracellular and external signals such as environmental cues or nutritional insufficiency through this transcription-controlling process. According to the GO annotation, the molecular function category showed twenty-one GO terminologies; mostly indicated to several enzymatic functions, and the others related to protein binding (Fig 2D). Here, the DNA and protein

**Table 1. Hypothetical proteins functionally annotated from the *B. paralicheniformis* strain Bac84.**

| No. | HP ID | Inferred function |
|-----|-------|-------------------|
| 1 | WP_158700706.1 | Metal-dependent hydrolase |
| 2 | WP_230368348.1 | Catalytic core DNA breaking-rejoining enzymes |
| 3 | WP_095290960.1 | RNA polymerase sporulation sigma factor SigK |
| 4 | WP_026579962.1 | YhzD-like protein |
| 5 | WP_224146215.1 | Response regulator aspartate phosphatase |
| 6 | WP_095291534.1 | The YqzH-like protein family |
| 7 | WP_003179940.1 | The YgaB-like protein family |
| 8 | WP_020449960.1 | Inner membrane protein YiaA-like |
| 9 | WP_105981192.1 | YqaH-like protein |
| 10 | WP_020453622.1 | Bacteriophage A118-like, holin |
| 11 | WP_006638778.1 | Metal-responsive transcriptional regulator |
| 12 | WP_003180123.1 | Sigma-M inhibitor protein YhdK |
| 13 | WP_025810847.1 | Streptogramin lyase |
| 14 | WP_020450411.1 | RlpA-like domain superfamily |
| 15 | WP_105980832.1 | Phenylalanyl-tRNA synthetase |
| 16 | WP_009328837.1 | Flavin-phosphopantothenoylcysteine decarboxylase/Flavin prenyltransferase |
| 17 | WP_003180732.1 | Pathogenicity locus—Putative mitomycin resistance proteins |
| 18 | WP_199792123.1 | YetA-like protein |
| 19 | WP_020451108.1 | ESAT-6-like superfamily |
| 20 | WP_020451191.1 | YkyB-like protein |
| 21 | WP_026579751.1 | Transcription regulator DksA-related |
| 22 | WP_105980957.1 | Nudix_Hydrolase super family |
| 23 | WP_023857538.1 | YhzD-like protein |
| 24 | WP_020451915.1 | Heat Shock protein (Hsp20 proteins) |
| 25 | WP_020452052.1 | HesB-like domain superfamily |
| 26 | WP_026579290.1 | YqfQ-like protein |
| 27 | WP_020452371.1 | RmlC-like cupin superfamily |
| 28 | WP_234026546.1 | Chromosome segregation protein SMC |
| 29 | WP_023855527.1 | Response regulator aspartate phosphatase |
| 30 | WP_105981186.1 | Putative phage metallopeptidase |
| 31 | WP_105981199.1 | Alpha/Beta hydrolase fold |
| 32 | WP_003185659.1 | Swarming motility protein SwrA |
| 33 | WP_023857076.1 | Acyl-CoA N-acyltransferase |
| 34 | WP_023856950.1 | BslA (Biofilm surface layer A) |
| 35 | WP_026580354.1 | Immunity protein WapI-like/YxiJ super family |
| 36 | WP_023856884.1 | Six-hairpin glycosidase superfamily |
| 37 | WP_020453535.1 | Prephenate dehydratase |

interactions (sequence-specific and sequence non-specific binding) are involved in many biological processes including regulation of transcription, DNA repair, DNA modification, etc. [54]. Additionally, the proteins with enzymatic functions have potential biotechnological applications [55, 56].

Additionally, 15 HPs carried homologous sequences with described functions were found in BlastP analysis whereas the remaining HPs were matched to uncharacterized family proteins and/or hypothetical proteins (S6 Table). All the 15 HPs that matched with functional proteins in the BlastP analysis were functionally similar to the anticipated functions. We also analyzed the promoter regions of all 37 proteins. Promoter segments are required for the start

of transcription at a certain genomic site. Several conserved regions such as the Pribnow box and -35 box were determined along with the SD sequence (S2 Fig). These conserved sequences are vital for the binding of RNA polymerase and ribosome [57, 58]. The SD-sequence initiates the translation process and has a huge influence on protein expression levels [59, 60]. It was found that all 37 proteins have SD sequences. The findings from the promoter analysis of the 37 proteins indicate that further experimental validation is worth pursuing. We did not find any study regarding the experimental transcription data sets of the organism.

Furthermore, the DEG database was utilized to predict fundamental genes (S7 Table). This database adapts both in vitro and in vivo experiments to detect fundamental genes which are essential for cellular machinery [38]. Though different challenging lab experiments were used to detect the essential genes such as RNA interference, gene knockouts, and transposon mutagenesis [61], this DEG database offers an alternative for predicting essential genes. In our analysis, we did not find any essential genes among the targeted 37 HPs.

## Physicochemical characterization and subcellular localization

To evaluate the physicochemical characteristics and their cellular distribution the sequences of the screened 37 HPs were used (S8 Table). Most of the studied proteins had molecular weight (MW) values over 10000 Da. Proteins with a lower MW ($< 10000$ Da) need special modifications for analysis in the SDS-PAGE system [62]. Hence, the first few HPs with lower MW require special attention to perform further lab experiments. The pH value of a protein at which it carries no net electrical charge is known as isoelectric point pI. For our selected HPs, it ranged from 4.4 to 10.48 and 11 proteins have acidic nature (pI $< 7$), whereas others were found to be basic. Along with the MW, the pI also helps in the laboratory analysis of proteins [63].

The aliphatic index (AI) is used to evaluate the protein thermostability and our HPs were in the range of 55.19–145.1. The range of temperatures at which a protein will be stable increases with increasing AI values [64]. Protein WP_003180123.1, associated with growth and survival after salt stress showed the highest value of 145.1. The instability index (II) was applied to get the idea regarding in vitro protein stability. 15 HPs were considered to be unstable, and 22 HPs were stable. The cut-off values $>40$ and $<40$ were used to categorize stable and unstable proteins, respectively [65]. The GRAVY indicates the interactive nature of a protein with water [66]. Among these 37 HPs, only four (WP_158700706.1; WP_003180123.1; WP_023857538.1 and WP_020453535.1) showed positive values which indicates that these might be hydrophobic.

Moreover, the cellular localization of proteins is vital for their biological functions in a specific environment [67–69]. Among the 37 HPs, most of the proteins were determined as cytoplasmic. Several cytoplasmic proteins are in the regulation of several functional processes including biosynthesis, regulatory activities, and transport which may help environmental bacteria to compete with the neighboring organisms in the same ecological niche [70]. Additionally, we only found 4 proteins to have signal peptides that are critically related to protein secretion [71].

## Protein-protein interactions

To determine the interaction partners of the HPs, we performed a protein-protein interaction analysis [72]. In this study, protein WP_095290960.1, RNA polymerase sporulation sigma factor SigK showed a very strong interaction (score 0.930) with the sporulation stage IV protein A (SpoIVA) which is involved in sporulation [73]. WP_006638778.1 interacted with EndoA–a putative RNase (score 0.988) with functional endoribonuclease activity [74]. WP_009328837.1

was found to interact with the YacB (score 0.987) which catalyzes the phosphorylation of pan-tothenate [75]. The protein WP_023855527.1 showed interaction with the Raca protein which is required for the formation of axial filaments [76]. All these findings along with the other predictions (S9 Table and S2 Fig) strengthened our functional predictions.

## Tertiary structure predictions

X-ray crystallography has become a robust approach to determining novel protein structures [77]. The functional annotation methods in combination with the protein structure analysis are evident to lead to the interpretation of uncharacterized proteins [78, 79]. In this study, we employed the protein structure homology-modeling server SWISS-MODEL to have the tertiary structures and used the UCSF Chimera software to visualize the models. Next, we compared the structures of known proteins with the modeled structures to check the degree of similarity (Fig 3A & 3B).

We successfully build the three-dimensional models for 9 HPs with identity above 30% and the details were listed in the S10 Table. We also checked the quality of the models with the Ramachandran plots and scores (S10 Table and S3 Fig). Structural comparisons were performed based on the Needleman-Wunsch algorithm [80]. We observed different percentages of structural similarities between the models and known proteins (S10 Table). The alignment results from the structural comparisons were shown in S4 Fig. The structural data collected for several HPs has validated the precise functional annotation. For instance, WP_105981199.1 and WP_023856950.1 showed high identities and resolutions which were functionally annotated as Alpha/Beta hydrolase and BslA (Biofilm surface layer A) respectively. The structures built for these two proteins were determined by X-ray crystallography from two *Bacillus sp.* and those two template proteins have similar functions as we predicted in this study. In this way, proteins with similar sequences usually exhibit similar functions. Proteins dissimilar to current PDB entries may correspond to novel functions. In addition, several final protein models were visualized using the Chimera 1.16 and compared to the predicted models suggested by AlphaFold (S5 Fig). We used the AlphaFold since Alpha-Fold has been demonstrated to be more accurate than Nuclear magnetic resonance spectroscopy (NMR) [81]. The findings showed similarities among the predicted models by Swiss-Model vs AlphaFold.

## ROC performance measurement

The availability of genome sequences is increasing which is also allowing more scope to do the computational protein analysis. As these analysis methods are solely dependent on autonomic computing, the accuracy of these methods should be high. The ROC analysis is a broadly applied technique for evaluating the tool's accuracy. The employed pipeline had an average accuracy of 98 percent (Table 2), and the ROC analysis's findings supported the strong dependability of the tools used.

**Table 2. ROC results of the tools used in this study.**

| Software | Accuracy (%) | Sensitivity (%) | Specificity (%) | ROC area |
|---|---|---|---|---|
| Pfam | 99.0 | 98.0 | 100 | 0.99 |
| InterPro | 100.0 | 100.0 | 100.0 | 1 |
| CATH | 100.0 | 100.0 | 100.0 | 1 |
| SUPERFAMILY | 96.0 | 94.7 | 100.0 | 0.99 |
| SCANPROSITE | 97.0 | 93.8 | 100.0 | 0.99 |
| SMART | 98.0 | 97.0 | 100.0 | 1 |
| CDD-BLAST | 96.0 | 65.9 | 100.0 | 0.985 |

## Proteins with biotechnological potentials

We found several proteins that can be used for biotechnological applications.

WP_158700706.1 was predicted as a Metallo-dependent hydrolase (the amidohydrolase superfamily). This group includes numerous hydrolytic enzymes with a varied spectrum of substrates and reactions. The microbial obtained amidohydrolase possesses extensive biotechnological applications that include cosmetics, food, and therapeutics, especially as an anticancer/anti-proliferative agent [82, 83]. This hydrolase group also contains amylases and α-amylase derived from *B. licheniformis*, *B. amyloliquefaciens* and *B. stearothermophilus* which has been commercially used in fermentation, paper, and textiles industries [84, 85].

Protein WP_020453622.1 is a Bacteriophage A118-like, holin that involves the lysis of bacterial membrane [86]. These holins can be utilized for controlled pore formation and can promote the release of the desired products. Microorganisms are used and improved for the industrial manufacture of a wide range of substances, including pharmaceuticals and biofuels. These target compounds can be sequestered inside the cell causing toxic effects to the chassis without an efficient active efflux system. In this case, Holin-mediated cell lysis offers an efficient releasing mechanism [87]. One of the rate-limiting steps is releasing products from the microbial host for biotechnology-based chemical production on an industrial scale. Holins can provide an affordable and effective method of product release in many instances where the use of mechanical disruption or solvent extraction increases the cost of production [88]. Liu and Curtiss applied phage holin/endolysin cassettes containing a nickel-inducible signal transduction system into the chromosome of *Synechocystis sp*. strain PCC6803 which is being developed for biofuel production [89]. They successfully eliminated the chemical or mechanical removal step by just adding nickel to the culture medium resulting in cell lysis. Another group utilized a light-inducible lytic mechanism in the same cyanobacterium for similar purposes [90].

The protein WP_009328837.1 was predicted as Flavin-containing phosphopantothenoyl-cysteine decarboxylase which is involved in coenzyme A (CoA) biosynthesis [91]. CoA is a crucial cofactor involved in many metabolic processes including secondary metabolites production. These distinctive features make CoA an economically significant chemical compound in the cosmetic, and therapeutic industries [92]. Hence, the catalytic abilities of this enzyme make it of immense biotechnological significance.

The protein WP_020452371.1 is in the RmlC-like cupin superfamily and RmlC is a dTDP-sugar isomerase enzyme (dTDP—deoxythymidine diphosphates). This enzyme is involved in the L-rhamnose synthesis, commonly found in bacteria and plants [93, 94]. This sugar getting more interest due to its wide range of substrate specificity and its excellent potential for various unique sugars syntheses such as D-allose, D-cellulose, L-mannose, L rhamnulose, L-spotose, and L-talose [95]. Besides, rhamnose is combined with lipids to form rhamnolipids that can be used as potential biosurfactants [94].

The protein WP_105981199.1 contains an α/β-hydrolase fold that includes proteases, lipases, peroxidases, esterase, epoxide hydrolases, dehalogenases, and many others [96]. Therefore, this protein can be studied further to uncover its actual functionality as several hydrolases are being used in industrial processes [56]. Additionally, an α/β-hydrolase fold protein was also studied which is involved in the cyclic oligopeptide antibiotic 'thiostrepton' biosynthesis [97].

The protein WP_023857076.1 carries a structural domain found in numerous acyl-CoA acyltransferases including the N-acetyl transferase (NAT) [98]. Several NATs from *Bacillus sp*. Have shown the capability to metabolize xenobiotic compounds that are highly toxic contaminants of groundwater and soils [99]. This study showed that a class of industrial contaminants

or by-products of agrochemicals named "Arylamines" can be converted into less toxic states by *Bacillus* NATs. Hence, our WP_023857076.1 protein should be studied further to find out its bioremediation potential. Additionally, a synthetic N-acetyltransferase (MAT—methionine sulfone N-acetyltransferase) from a bacterial source was utilized to successfully design herbicide "Phosphinothricin" -resistant rice and Arabidopsis [100].

Different glycosyltransferases transfer sugar parts from donor molecules to acceptors to form glycosidic bonds and involve in disaccharides, oligosaccharides, and polysaccharides biosynthesis. Several microbial glycosyltransferases are frequently applied in food processes such as in the shelf-life improvement of bakeries, production of glucose, fructose, or dextrins, lactose hydrolysis, food pectins modification, and many others [101, 102]. In our study, protein WP_023856884.1 has the catalytic domain of the Six-hairpin glycosidase superfamily. To use this class of enzymes in different industrial conditions several enzymes functional in alkaline/ acidic pH and/or at high temperatures have been discovered from various microorganisms [103–105]. In several studies, bacterial glycosidases were characterized to improve human health and the treatment of different diseases [106, 107].

The WP_020453535.1 was anticipated to be a prephenate dehydratase that is involved in the biosynthesis of phenylalanine and phenylalanine is an essential amino acid for animals. Recently, the interest in microbial production of L- phenylalanine has increased [108]. It has been widely used in food and feeds as a taste and aroma enhancer, in pharmaceuticals as the drug's building block, as well as used in cosmetics as an ingredient [109, 110].

## Proteins with adaptational functions to extreme environments

In this study, we identified 12 HPs that may have a significant role for *B. paralicheniformis* in the adaptation to extreme environments.

Sporulation aids bacterial survival in extreme environments by limiting active growth [111]. We found protein WP_095290960.1 as RNA polymerase sporulation sigma factor SigK which is involved in the gene expression controlling during sporulation [112]. Two HPs (WP_224146215.1 and WP_023855527.1) were identified to be the aspartate phosphatase, which regulates the phosphorelay for sporulation initiation by dephosphorylating Spo0F-P [113]. In this way, these HPs can be predicted to play crucial roles in adaption, and survival in extreme environments.

The protein WP_006638778.1 is a metal-responsive transcriptional regulator which can be engaged in the homeostasis and metabolism of any specific metal. These metal-responsive transcriptional regulators allow mechanisms for selective metal ion accumulation and utilization as well as tightly regulate intracellular metal trafficking mechanisms [114]. Metals can be limited in the environment or can be in high amounts that cause toxicity in extreme environments. Hence, a metal-responsive transcriptional regulator protein might be essential to the microorganism for the evolution and adaptation in that specific extreme environment [115]. Likewise, WP_026579751.1 is related to the transcription regulator DksA. It is an RNA polymerase-binding transcription factor and is involved in different stress conditions, including nitrosative stress, nutritional shortage, and other environmental stresses [116, 117]. So, this HP can be taken part in extreme environmental adaptations.

We detected a sigma-M inhibitor protein (WP_003180123.1). The sigma-M (YhdM) gene is essential for growth and survival in salt stress conditions [118]. Our predicted Sigma-M inhibitor WP_003180123.1 might play role in salt stress adaptation similarly to a previous study [119].

Protein WP_105980957.1 contains a Nudix hydrolase domain that hydrolyzes intracellular nucleotides, regulates their levels, and removes potentially toxic derivatives [120]. Some

superfamily members can degrade mutagenic, oxidized, and damaged nucleotides that may occur due to exposure to extreme environments [121].

As mentioned earlier, WP_023857076.1 carries a structural domain found in numerous acyl-CoA acyltransferases including- GCN5-related N-acetyltransferases (GNAT) and Glycine N-acyltransferase [122]. The proteins from these classes were studied and found to be involved in the adaptation to diverse environmental stress conditions including high salinity, pH tolerance, nutrient stress, etc. [123, 124].

Small Heat shock proteins are abundant molecular chaperones that counteract the aggregation of protein upon stress-induced unfolding [125]. We identified protein WP_020451915.1 as a heat shock protein (Hsp20). Several studies showed that Hsp20 responds to different environmental stresses including severe heat, hydrogen peroxide, desiccation, and osmotic shocks [126–129]. Therefore, WP_020451915.1 might have adaptational functions to extreme environments.

The HesB-like domain is observed in several microbial nitrogen fixation proteins that are associated with FeS-cluster assembly [130]. Previous studies found that proteins having a HesB-like domain are involved in different metal resistance and thermal stress conditions [131, 132]. HesB-like domain-containing protein WP_020452052.1 might also play role in survival in the extreme environment specifically in metal-rich or metal deficient conditions.

The WP_003185659.1 protein was identified as a swarming motility protein SwrA which is a transcription factor. It drives the *fla/che* operon, which encodes the components of the flagella, and causes swarming motility [133]. Another study showed that SwrA is involved in bacterial motility [134] and bacterial motility might be significant in extreme temperatures [135].

The WP_023856950.1 protein was predicted as a biofilm surface layer A (BslA) protein which acts as a hydrophobin and participates in biofilm assembly [136]. Certain microorganisms have great resistance to environmental challenges because of biofilm development [137–139]. Therefore, this protein might be crucial for adaptation to harsh environments.

## Conclusions

Protein macromolecules are involved in numerous biological processes. Hence, functional annotation of proteins is crucial. An in silico approach was employed in this study to attribute functional annotation of HPs from the *B. paralicheniformis* strain Bac84 genome. We predicted the functions of 37 HPs from this bacterium. The determination of physicochemical parameters and subcellular localization were effective to understand the specific properties of the annotated proteins. The PPI and tertiary structures of these proteins were also explored which assisted to obtain more understanding of the annotated proteins. Several protein structures were also validated by the AlphaFold protein modeling. We identified several proteins with biotechnological potentials as well as proteins having the possibility to be involved in extreme environmental adaptation of the *B. paralicheniformis* strain Bac84. Moreover, the findings of this strategy suggested that it can be utilized to perform the predictive annotations of unknown proteins. The combination of such in-silico analysis with the proper lab experiments was successful to obtain functional annotations of HPs from different organisms [140–142]. Furthermore, the results also open prospects for further research of this bacterium for biotechnological applications.

## Supporting information

**S1 Fig. Protein-protein interaction networks obtained from STRING analysis.** Networks are visualized using Cytoscape.
(PDF)

**S2 Fig. Promoter analysis of the 37 proteins using BPROM.**
(PDF)

**S3 Fig. Ramachandran plots for the 3D models of the 9 proteins by the SWISS-MODEL serve.**
(PDF)

**S4 Fig. Alignment results from the superposition analysis.**
(PDF)

**S5 Fig. Comparison of the structures predicted by AlphaFold and Swiss-Model.**
(PDF)

**S1 Table. All the hypothetical proteins from the *B. paralicheniformis* strain Bac84.**
(XLSX)

**S2 Table. List of bioinformatics tools and databases used.**
(XLSX)

**S3 Table. Annotation dataset results for the 414 hypothetical proteins submitted to the workflow with Pfam, InterPro, CATH, SUPERFAMILY, SCANPROSITE, SMART, and CDD-Blast.**
(XLSX)

**S4 Table. List of selected HPs from the *B. paralicheniformis* strain Bac84.**
(XLSX)

**S5 Table. GO terms by Argot2.5 for all the HPs.**
(XLSX)

**S6 Table. Results of the BlastP search for similar sequences against the non-redundant (nr) database.**
(XLSX)

**S7 Table. Result of essential gene prediction using DEG database.**
(XLSX)

**S8 Table. List of predicted physicochemical parameters, sub-cellular localization, and prediction of transmembrane helices for the selected 37 HPs.**
(XLSX)

**S9 Table. Protein-protein interactions analyses of the 37 HPs.**
(XLSX)

**S10 Table. Tertiary structural information of HPs from *B. Paralicheniformis* strain Bac84.**
(XLSX)

**S11 Table. Dataset of functional annotation for 100 functionally known proteins from *B. paralicheniformis* strain Bac84 using the same pipeline used for the HP prediction.**
(XLSX)

## Acknowledgments

We thank Research Square for making our publication available online as a preprint.

## Author Contributions

**Conceptualization:** Md. Atikur Rahman, Uzma Habiba Heme, Md. Anowar Khasru Parvez.

**Data curation:** Md. Atikur Rahman, Uzma Habiba Heme.

**Formal analysis:** Md. Atikur Rahman, Uzma Habiba Heme, Md. Anowar Khasru Parvez.

**Investigation:** Md. Atikur Rahman, Uzma Habiba Heme.

**Methodology:** Md. Atikur Rahman, Uzma Habiba Heme, Md. Anowar Khasru Parvez.

**Project administration:** Md. Atikur Rahman, Md. Anowar Khasru Parvez.

**Resources:** Md. Anowar Khasru Parvez.

**Software:** Md. Atikur Rahman, Md. Anowar Khasru Parvez.

**Supervision:** Md. Anowar Khasru Parvez.

**Validation:** Md. Atikur Rahman, Md. Anowar Khasru Parvez.

**Visualization:** Md. Atikur Rahman, Uzma Habiba Heme, Md. Anowar Khasru Parvez.

**Writing – original draft:** Md. Atikur Rahman.

**Writing – review & editing:** Md. Atikur Rahman, Uzma Habiba Heme, Md. Anowar Khasru Parvez.

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
