## [Decision Letter · Decision Letter 0]

7 Sep 2022

PONE-D-22-21684In silico functional annotation of hypothetical proteins from the *Bacillus paralicheniformis* strain bac84 reveals proteins with biotechnological potentials and adaptational functions to extreme environmentsPLOS ONE

Dear Dr. A.K. Parvez

Thank you for submitting your manuscript to PLOS ONE. After careful consideration, we feel that it has merit but does not fully meet PLOS ONE’s publication criteria as it currently stands. Therefore, we invite you to submit a revised version of the manuscript that addresses the points raised during the review process. While the reviewers were positive about your manuscript, they also raised some concerns and suggestions that can improve its scientific impact. During your revision, please, consider all the Reviewer’s comments, but, please, pay particular attention to those indicating that, The quality of all the figures and tables, as well as the figure legends require to be significantly improved; Additional bioinformatic approaches are required to corroborate the structure and predicted function of the modeled proteins from Figure 3; Experimental and bioinformatic approaches must be implemented to corroborate if the genes encoding the predicted 37 proteins, a) possess transcriptional *in cis* elements and, b) if they are transcribed; Several conclusions in the manuscript are not supported by the bioinformatic evidence presented, therefore, these conclusions must be properly adjusted.

We look forward to receiving your revised manuscript.

Kind regards,

Mario Pedraza-Reyes, Ph.D.

Academic Editor

PLOS ONE

**Comments to the Author**

1. Is the manuscript technically sound, and do the data support the conclusions?

Reviewer #1: Partly

Reviewer #2: Yes

Reviewer #3: Yes

2. Has the statistical analysis been performed appropriately and rigorously? 

Reviewer #1: N/A

Reviewer #2: Yes

Reviewer #3: N/A

3. Have the authors made all data underlying the findings in their manuscript fully available?

Reviewer #1: Yes

Reviewer #2: Yes

Reviewer #3: Yes

4. Is the manuscript presented in an intelligible fashion and written in standard English?

Reviewer #1: No

Reviewer #2: Yes

Reviewer #3: Yes

5. Review Comments to the Author

Reviewer #1: The work presented by Rahman and colleagues provides an interesting approach for annotating hypothetical proteins in the genome of Bacillus paralicheniformis strain bac84. The use of well-established tools is interesting and relevant. The findings, this reviewer agrees that they have both basic biology and biotechnology interests. The manuscript has some issues in the writing that a thorough revision can fix. Overall, this reviewer thinks that the approach is sound and the results interesting. However, in the following lines, I respectfully provide some concerns regarding how the results are shown and described in the manuscript. I hope they are useful for the authors and may help assess important aspects of the study.

Overall, the approach reported here is sound, using well-established tools that are valid and relevant for assessing sequence features that may stand out, and thus, assessing the functional role of proteins of unknown function is important. The main concern of this reviewer regarding the manuscript is that the discussion is highly speculative regarding some hypothetical proteins annotated in this study. The discussion can be focused on searching which genes contain a clearer homolog along with good structural comparisons (like structural alignments, authors can either generate the model or use those available in the AlphaFold2 database at EBI) to assess the extent of both homology and relevant residues/domains for each hypothetical protein. This strengthens the workflow used in this manuscript. This reviewer suggests using the metabolic enzymes found, for example, WP_020453535.1, and comparing it with other similar enzymes. Using structural alignments, authors may confirm the fold and critical catalytic residues, thus confirming the success of the analysis.

Additionally, with my previous comment, I suggest adding in the introduction or in the discussion some features of this particular strain, especially those regarding the known phenotypes it displays, such as biofilm formation, resistance or sensitivity to antibiotics or heavy metals and all that is known about its physiology. Please, check all the available literature supporting the possible role of these hypothetical proteins in the physiology of this strain.

By checking the hypothetical proteins in Supplementary Table 1, why authors did not discard very short sequences? Can these be annotation artifacts in this and other genomes of close species? Is there any suggestion that these proteins may be real ORFs such as promoter or regulatory sequences? This study will greatly benefit from including the promoter analysis of the 37proteins with a final annotation and support that further experimental validation is worth pursuing. This reviewer also thinks lines 201-202 are unnecessary; why these small proteins should gain focused attention?

In lines 217-220, the authors make a statement that is hard to follow in the figures presented in this manuscript. I kindly request to provide a figure with the genes involved in response to environmental cues. Also, I recommend toning this statement down since no experimental validation is provided for the hypothetical proteins analyzed here.

Several hits (Table S6) seem to be related to phage integrases. Are these proteins in integrons or remanent phages? If they are, this reviewer thinks that additional analysis should be conducted, such as G+C bias with the rest of the genome and codon usage (CAI) analysis to verify if these genes are horizontally acquired and perhaps further analyze them in specific phage databases for enzymatic or structural features more associated with phages.

In the conclusion, I also think that the statement that "… this strategy provided us with excellent results…" is too optimistic. Of 414 proteins, 37 are just a fraction of the unknown proteins. I kindly invite the authors to tone down the manuscript.

In the tertiary structure prediction section, Swiss-model is a powerful tool, but for some time now, there have been open, collaborative notebooks for using AlphaFold2. In this reviewer's opinion, small proteins should be de novo modelled with AlphaFold2 since recently has been shown to be as effective as NMR structural determinations (Structure. 2022 Jul 7;30(7):925-933.e2. doi: 10.1016/j.str.2022.04.005.). The use of template-based models is very useful for the objective of this manuscript. This recommendation is to validate the structure of at least some of these proteins to prevent bias towards known structures. If the two models are very similar, then confidence in the putative function is stronger. Otherwise, this may result from a forceful prediction of a known template.

Regarding table 2, the average is not advised here since each tool uses different algorithms to predict and assess conserved domains or features in these proteins. Each tool has its own performance; therefore, the table is informative, but I suggest removing the average, which may be misleading.

The general quality of the main figures is low; they look a bit fuzzy; I recommend using higher resolution images. I think this may have been the result of the generation of the review pdf, but I kindly suggest revising this.

Figure 2 legend is poorly described. I recommend indicating the neighborhood of the found proteins and exploiting the data shown here in more detail. Also, Figure 3 provides little information. This reviewer suggests using structural comparisons (alignments) to support their findings further, as mentioned above.

Minor comments:

This reviewer suggests modifying the use of "functional annotation" throughout the manuscript. I think the authors should tone down the manuscript since no experimental evidence of these genes is provided. As stated above, I believe this is a good starting point for characterizing these genes. My humble suggestion is to use "predictive annotation."

Line 2, capitalize Bacillus, please.

Line 42-43, please italicize B. licheniformis

Line 44, please change "things" to "products"

Line 66, please change expressional for expression and function associated data.

In line 61, please change "considerable" with the corresponding percentage of the total coding capacity of this strain.

Line 96, please change to Table S3. Also, please revise the numbering of the Supplementary Tables; S11 is S10.

Line 102, please capitalize fasta

Line 114, please correct Line 114, "of a helps to" is not correct.

Line 127, please correct to Fig S1. Also, I suggest changing the labels for supplementary Tables throughout the manuscript.

Please clarify line 190; RNA interference usually is an experimental approach for eukaryotic organisms.

Line 226, I suggest the following modification "…putative RNase (score 0.987) with functional endoribonuclease activity"

Please check gene and protein names to use the correct format; some lack capital letters at the beginning of the name, and genes lack italics.

Line 257, in other places, authors use this statement. I suggest changing to "can be used for biotechnological applications" in the manuscript.

I suggest removing line 280, which is repetitive with this paragraph.

In line 281, I think "anticipated" is not a correct word here, perhaps predicted.

Please check the grammar in line 328; this line is confusing.

Please add italics in line 367

I hope these comments are useful for the authors.

Best regards

Reviewer #2: This manuscript employed a structured in-silico approach incorporating numerous bioinformatics tools and databases for functional annotation of hypothetical proteins from Bacillus paralicheniformis Bac84 reveals proteins with biotechnological potentials and adaptational functions to extreme environments. The knowledge of these hypothetical proteins' potential functions aids B. paralicheniformis Bac84 in effectively creating a new biotechnological target and also facilitate a better understanding of the survival mechanisms in harsh environmental conditions. The findings are meaningful and my comments are as follows:

1. Functional mining of these hypothetical proteins is interesting in B. paralicheniformis Bac84. Are all these 37 hypothetical proteins existing in the species B. paralicheniformis? Which of these hypothetical proteins are specific in strain Bac84? Are these 37 hypothetical proteins can be used as indicators for the classification and identification of the two species B. paralicheniformis and B. licheniformis? I think the authors could add these contents into the Results or Discussion parts of this manuscript if possible.

2. I wonder if the authors considered whether these hypothetical proteins can actually be expressed? RNA-seq or RT-PCR technologies could be applied to further understand the expression patterns of these hypothetical proteins.

3. The English writing of this manuscript could be further improved. Some small errors should be changed, for example, in line 2 and 6, “bacillus” should be changed to “Bacillus”; in line 37, the word “strain” should be standardized; in line 42 to line 43, “B. licheniformis” should be changed to “B. licheniformis”.

Reviewer #3: Atikur Rahman et al., analyzed in silico 414 sequences of hypothetical proteins from Bacillus paralicheniformis strain bac84 with the aim of attributing, at least in a predictive form, its functional role based on sequence and structure homology with well-known proteins. 37 hypothetical proteins resulted with an associated function before the bioinformatic analyses using several online tools. The authors also mentioned in this manuscript the potential of their findings to possibly use some of these proteins in biotechnological applications. This draft could be recommended for publication in PLoS ONE not without first addressing several points of concern detected.

A) In figure 1, outlined green boxes are depicted to indicate several steps performed as part of the general protocol applied to all sequences, however, it is not clear why some boxes were green outlined and not others. I assume that tasks described in the green outlined boxes were only applied if the HPs analyzed fulfilled the same predicted function in at least three bioinformatics tools. This is an elementary observation but it could be confusing when interpreting the figure. I suggest that a figure caption with information that allows understanding of the content of the figure be added since the one described in line 99 is only limited to giving the figure a name but does not explain its graphic content.

B) Results of gene ontology shown in figure 2 are confusing to my eyes. What is the rationale to use the “bubbles” representation? What X axis indicated? If the distances between the bubbles do not indicate something significant and it is only to increase the area of the bubble based on the number of proteins that have a certain predicted function, I suggest that the data be represented in another way, for example, in a bar graph (as in section A), a pie chart or even in tables. This applies to items B and C of this figure.

C) Figure 3 is pretty poor. It is not very informative as it is only limited to showing the structures of the eleven modeled proteins and does not provide relevant information about the findings. If the reader wishes to know what theoretical function each protein has, and the whose predicted structure is shown, it is necessary to check the supplementary table S10. The latter is not at all practical when reading the manuscript. As in the other captions of figures 1 and 2, the caption of figure 3 is very insignificant because it does not provide information on the structures, how they were modeled, the selection of colors, and the reliability of the models presented, among others. Please consider these aspects to improve this figure.

D) Lines 180-181: Please name at least three types of DNA-Protein interactions in the biological context.

E) Line 230: S2 Fig must be S1.

F) In lines 244-245 the authors claim: “…proteins with similar sequences usually exhibit similar functions. Proteins dissimilar to current PDB entries may correspond to novel functions.” Can the authors do a more detailed analysis by comparing the structures of known (crystallized) proteins with the structures they modeled? Thus, it would be possible to have a clearer idea of how similar the three-dimensional structures are, as it is mentioned that a part of the modeled protein may resemble the already known one, but another part may not.

G) Lines 246-247: Can the authors be more specific about what they consider an "excellent degree of reliability? Table S10 shows the percentage of a favored Ramachandran structure, however, it gives little information on specific Ramachandran conformations for helices and B-strands. A supplemental figure showing each Ramachandran plot for at least the 11 modeled protein structures of Figure 3 might be beneficial.

H) Lines 248-254: “ROC” is not defined in the manuscript. Please define it. PLoS One is not a bioinformatic specialized journal, therefore, some terms may be unusual for different readers.

I) Line 263: a connector is necessary at the end of the line.

J) Do the authors consider it really relevant for the study to present the information in Table 2? Bioinformatic approaches used are all previously verified and properly referenced here. I suggest eliminating or including it as supplementary material.

K) Figure S1 shows the protein-protein interactions that resulted in the analysis performed on STRING, however, it is not specified what type of hypothetical or experimentally verified interaction was found. STRING provides information in a color code depending on the type of interaction (hypothetical, known, direct interactions, expression, operon array, among others), however, this information was omitted for the results shown using Cytoscape. I encourage authors to consider mentioning this information.

6. PLOS authors have the option to publish the peer review history of their article (what does this mean?). If published, this will include your full peer review and any attached files.

Reviewer #1: **Yes: **Bernardo Franco

Reviewer #2: No

Reviewer #3: No

---

## [Author Response · Author response to Decision Letter 0]

20 Sep 2022

We have added the file "Response to Reviewers". We have carefully checked all the comments and tried our best to address every one of them in the revision. We hope the manuscript after careful revisions meet your high standards.

---

## [Editor Report · Decision Letter 1]

28 Sep 2022

In silico functional annotation of hypothetical proteins from the Bacillus paralicheniformis strain bac84 reveals proteins with biotechnological potentials and adaptational functions to extreme environments

PONE-D-22-21684R1

Dear Dr. Parvez

We’re pleased to inform you that your manuscript has been judged scientifically suitable for publication and will be formally accepted for publication once it meets all outstanding technical requirements.

Kind regards,

Mario Pedraza-Reyes, Ph.D.

Academic Editor

PLOS ONE

---

## [Editor Report · Acceptance letter]

4 Oct 2022

PONE-D-22-21684R1 

In silico functional annotation of hypothetical proteins from the *Bacillus paralicheniformis* strain Bac84 reveals proteins with biotechnological potentials and adaptational functions to extreme environments 

Dear Dr. Parvez:

I'm pleased to inform you that your manuscript has been deemed suitable for publication in PLOS ONE. Congratulations! Your manuscript is now with our production department. 

Kind regards, 

on behalf of

Dr. Mario Pedraza-Reyes 

Academic Editor

PLOS ONE